# Interspecific Competition between Invasive *Spodoptera frugiperda* and Indigenous *Helicoverpa armigera* in Maize Fields of China

Yifei Song [1,2], Hui Li [2], Limei He [3], Haowen Zhang [2], Shengyuan Zhao [2], Xianming Yang [2] and Kongming Wu [2,*]

[1] Institute of Insect Sciences, College of Agriculture and Biotechnology, Zhejiang University, Hangzhou 310058, China

[2] State Key Laboratory for Biology of Plant Diseases and Insect Pests, Institute of Plant Protection, Chinese Academy of Agricultural Sciences, Beijing 100193, China

[3] Institute of Urban Agriculture, Chinese Academy of Agricultural Sciences, Chengdu 610000, China

* Correspondence: wukongming@caas.cn

**Abstract:** Since the fall armyworm (FAW) *Spodoptera frugiperda* invaded China, it has coexisted in maize fields with the native cotton bollworm (CBW) *Helicoverpa armigera*, but the population dynamics and competitive mechanisms between the two pests are not well understood. We evaluated interspecific competition between FAW and CBW by analyzing their bidirectional predation in the laboratory, survival rates when their larvae co-infested the same maize plant, and the population dynamics of both in the same maize field. In the predation tests, FAW and CBW larvae preyed on each other. However, the theoretical maximum predation of sixth-instar FAW larvae preying on first–second-instar CBW larvae was 71.4 and 32.3 individuals, respectively, while that of CBW was 38.5 and 28.6 individuals. The field co-infestation trials showed that the older larvae had a higher survival rate when the two pests co-infested the same maize plants, but young larval survival was higher for FAW than CBW. In the maize field from 2019 to 2021 in southern Yunnan, FAW populations were significantly higher than those of CBW. Our findings suggested that FAW larvae had a predation advantage over CBW, which might be an important reason for its dominance in Chinese maize fields. This result provides a scientific basis for developing a monitoring technology and for the integrated management of pests in invaded habitats of FAW.

**Keywords:** *Spodoptera frugiperda*; *Helicoverpa armigera*; interspecific interactions; predation function response; invasion ecology; maize

## 1. Introduction

The rapid development of global trade and climate change has created favorable conditions for invasion by alien species [1,2]. In China, the total economic losses caused by invasive species exceed USD 18.9 billion annually, with direct losses exceeding USD 7.7 billion [3,4]. In addition to directly damaging host plants, invasive pests can also affect native pests through interspecific competition when their niches overlap, and more competitive species may become dominant [5–8]. For example, the invasive stem borer *Chilo partellus* (Swinhoe) rapidly increased the share of the total borer population by terminating diapause earlier and shortening their life cycle compared to the native maize stalk borer *Busseola fusca* in Africa [9]. As an invasive species worldwide, the females of MEAM1 *Bemisia tabaci* only prefer to mate with homotypic males, while males can mate with native whiteflies. Through reproductive interference, MEAM1 gradually evolved into the dominant species in China [10]. Therefore, interspecific interaction with local pests might be one of the main factors determining whether invasive pests colonize a habitat and spread to new ones.

The fall armyworm (FAW), *Spodoptera frugiperda* (J. E. Smith) (Noctuidae), native to tropical and subtropical America, is an important invasive pest worldwide [11]. In 2018,

FAW invaded India, Southeast Asia (such as Myanmar and Vietnam), and Yunnan Province in China [12–15], serving as the source population for northern China, Japan, and South Korea [16,17]. Although FAW has been known to feed on 353 host plants across the globe, maize has become the most seriously affected in China, accounting for around 95% of the total area damaged by the pest in 2019 and 2020 [18–21].

The cotton bollworm (CBW), *Helicoverpa armigera* (Hübner) (Noctuidae), a polyphagous species with more than 200 host plant species, including maize, cotton, and wheat [22–24], also causes huge agricultural losses [25]. On maize, CBW and FAW mainly feed on tassels, ears, or kernels during the reproductive stage [18,26]. Both species can co-infest the same plant [27], but the dynamics and mechanisms of their competition are not well understood. A previous study showed that the intraguild predation ability has been proven to play important roles in the competitive advantage of FAW over indigenous *Spodoptera litura* in China [28]. In this paper, we analyzed bidirectional predation between FAW and CBW, and the larval survival rates on the same maize plant in a competition scenario. Finally, we also surveyed the population abundance in local maize fields.

## 2. Materials and Methods

### 2.1. Insects and Maize Plants

Late instar larvae of FAW were collected from maize fields in Mengmao (23.58° N, 97.48° E), Ruili City, Yunnan Province, in June 2020. CBW late instars were collected from maize fields in Langfang (39°51′ N, 116°60′ E), Hebei Province, in July 2020, and reared in March in Mengmao. Both FAW and CBW colonies were reared for around 1 year after collection in the field. Larvae of FAW and CBW were fed an artificial diet of wheat bran and soybean flour [29]. The 1st–2nd-instar larvae of both FAW and CBW were reared in 1000 mL plastic cages (length × width × height, 17 cm × 12 cm × 7 cm) with artificial diets. The 3rd–6th-instar FAW and CBW larvae were reared singly in glass tubes (25 mL). The FAW moths were reared in insect cages and CBW were placed in round plastic buckets covered with absorbent gauze and a disinfected wet towel. Adults were fed a 15% honey–water solution. All insects were maintained in several climate chambers (MGC–350, Changzhou Jintan Liangyou Yiqi Ltd., Jintan, China) at 25 °C, 60% RH with 14 h/10 h (light/dark).

Maize (Hanikesi SBS902, Xiamen Huatai Grain Seedling Co., Ltd., Xiamen, China) was planted in December 2021 in a single field of 330 m$^2$ in Mengmao (20 cm between plants in rows 40 cm apart). Maize plants (growing to the R1 stage) were taken as the insect's food source in the laboratory assays, and we conducted co-infestation trials in the field [30].

### 2.2. Predation Assays

First, the freshly molted larvae at different instar stages were collected at 9 o'clock and 21 o'clock every day and starved for 12 h separately in a Petri dish (diameter 8.5 cm, height 1.3 cm). Predation assays were performed on the various combinations of species × stage combinations (1st to 6th instar larvae of FAW and CBW). Pairs of larvae at different stages of each species were placed in sealed Petri dishes without any food (8.5 × 1.3 cm). For example, one 1st instar larva of FAW was paired with one 1st, 2nd, 3rd, 4th, 5th, or 6th instar larva of CBW in a sealed Petri dish. Thus, 36 pairing treatments were tested, with 30 replicate pairs for each treatment. Insects were incubated in climate chambers as described in the section on insects and maize plants. The number of alive or dead FAW and CBW larvae was recorded after 24 h, and the predation rate of FAW on CBW is as follows: the number of dead CBW/30 × 100% and vice versa. The laboratory experiment lasted for nearly two months.

### 2.3. Predation Functional Responses

On the basis of predation assays, the predation ability of the 6th-instar larvae of FAW and CBW was assessed by functional responses. One single 6th-instar larva of one species was chosen as the predator and placed with a varying density of 1st- and 2nd-instar larvae of the other species as the prey in a 500 mL transparent, plastic container (8.0 cm bottom

diameter $\times$ 7.0 cm height). The prey density for the 1st-instar larvae was 10, 20, 50, 100, and 200 individuals, and 10, 20, 40, 80, and 100 individuals for the 2nd instars. The freshly molted (<12-h) 6th-instar larvae predators were starved for 12 h before testing. Each plastic container was filled with fresh maize tassels to reduce larval cannibalism, and plastic containers containing a given number of prey and maize tassels without any predators served as the control. The containers were incubated as described above. The amount of alive prey was counted after 24 h. Each predator–prey combination (1 single 6th-instar larva predator vs. a varying number of 1st to 2nd-instar larvae of prey) was replicated five times.

### 2.4. Field Competition and Population Dynamics

### 2.4.1. Interspecific Competition between FAW and CBW on Maize Plants

In the maize field, as described in the section on insects and maize plants, 10 maize plants (R1 stage) were randomly selected for treatment, and all arthropods on the plants were removed with a brush before testing. On each plant, only one ear of maize was tested. We tested three co-infestation treatments: one 1st-instar FAW larva paired with one 1st- or 3rd-instar CBW larva, and one 1st-instar CBW larva paired with one 3rd-instar FAW larva. The paired pests were placed into the tassel on the same ear. The survival larvae of FAW and CBW were only counted after 5 days of destructive sampling. Each treatment included 3 replicates; the control treatments included only 1 1st-instar larva of FAW or CBW on maize tassels.

### 2.4.2. Population Density Survey of FAW and CBW in Local Maize Fields

From mid-November to early December 2019 to 2021, maize fields were surveyed for larval occurrence of FAW and CBW in Mengmao. The survey was carried out once the maize developed to VT–R1, and the number of surveyed fields was 20, 20, and 35 from 2019 to 2021, respectively; each field was visited one time. The number and development stage of FAW and CBW larvae on each maize plant were recorded. Within each field, a total of 500 maize plants were randomly selected along 5 points (100 plants per point). Maize fields were independently managed by local farmers spraying pesticides (0.13–0.27 ha per field; ~60,000 plants/ha) and planted with the maize hybrids 'Shuangse Xianmi' (Hebei Sanbei Seed Industry Co., Ltd., Chengde, China), 'SBS902' (Xiamen Huatai Wugu Seedling Co., Ltd., Xiamen, China), and 'Lvse Xianfeng F1' (Beijing Yanheyu Technology Development Co., Ltd., Beijing, China).

In December 2019–2021, most of the local maize has developed to the VT–R1 stage; FAW and CBW adults were captured using three light traps (metal halide lamp, model JLZ1000BT, Shanghai Yaming Lighting Co. Ltd., Shanghai, China) to determine population sizes [31]. The light traps were turned on at 19:30 and turned off at 06:30. The distance between each light trap was less than 500 m. Nylon net bags (60 meshes, 50 $\times$ 50 $\times$ 80 cm) were attached under the lamp trap, and FAW and CBW moths were counted every day.

### 2.5. Data Analyses

The predation rates between FAW and CBW larvae were analyzed using $\chi^2$ analyses. The results of co-infestation field trials, the larval occurrence of two pests in the fields, and the rapped number of adults were analyzed for differences among the treatments with a one-way analysis of variance (ANOVA), followed by Duncan's new multiple range test. The proportional data of the trials were arcsine-transformed to meet assumptions of normality and heteroscedasticity. The difference in the larvae abundance (number of larvae per 100 plants) and daily captured moths for FAW and CBW in each year was also analyzed using an ANOVA.

Data for predation functional responses were first subjected to a logistic regression using R version 2.0.1 (R Core Team 2017) to estimate the type of functional response [32], and they were then fit to a polynomial function as follows:

$$N_e/N_0 = \exp(P_0 + P_1N_0 + P_2N_0{}^2 + P_3N_0{}^3)/[1 + \exp(P_0 + P_1N_0 + P_2N_0{}^2 + P_3N_0{}^3)],$$

$N_e$ is the number of prey consumed, $N_0$ is the initial prey density, $P_0$ is the intercept, $P_1$ is the linear, $P_2$ is the quadratic, and $P_3$ is the cubic coefficients. In this trial, all prey numbers consumed ($N_e$) were corrected by subtracting the number of preys cannibalized in the controls. The predation functional response was determined as type II when $P_1 < 0$ and as type III when $P_1 > 0$ [33]. If $P_1$ did not differ significantly from 0, the trinomial was ignored until the coefficient was significant [32]. After the response type was determined, we used Holling's disc equation to calculate the parameters. For the type II response, the equation is:

$$N_e = aN_0T/(1 + aN_0T_h),$$

where $a$ is the attack rate, $T$ is the total available time for predation ($T = 1$ d in this experiment), and $T_h$ is the handling time of the predator. For the type III response, $a = (d + bN_0)/(1 + cN_0)$, and the equation is as follows:

$$N_e = (d + bN_0)N_0T/[1 + cN_0 + (d + bN_0)N_0T_h],$$

where $b$, $c$, and $d$ are the constants calculated using nonlinear least square regression with the nls function in R. The theoretical maximum predation value ($N_m = T/T_h$) was also calculated.

All analyses were conducted using SPSS version 23.0 (IBM, Armonk, NY, USA) or R version 2.0.1 (R Core Team 2017), and the cannibalistic number was calculated by larval survivors/total number in a plastic container.

## 3. Results

### 3.1. Predation Rates between FAW and CBW

Both FAW and CBW preyed on their counterparts in all larval combinations except for the first-instar larvae, and the predation rate increased significantly with the increasing life stage of the predator and decreased with the increasing life stage of the prey (Table 1). More specifically, the fifth- and sixth-instar FAW larvae preyed on all immature stages of CBW, with fifth- and sixth-instar FAW larvae consuming a respective 3.3% to 100% of CBW prey. However, second- and third-instar FAW larvae only preyed upon the same larval stage or younger CBW, consuming 10.0% to 76.7% of CBW prey. Similarly, second- to sixth-instar CBW larvae only preyed on the same stage or younger larvae of FAW, consuming a respective 3.3% to 100% of FAW prey. For the third–third, fourth–fourth, fifth–fifth, and sixth–sixth, the predation rate of FAW on CBW was 16.7%, 30.0%, 20.0%, and 16.7%, respectively, and 10.0%, 3.3%, 10.0%, and 6.7%, respectively, for CBW preying on FAW, indicating that FAW had a competitive advantage over CBW.

### 3.2. Predation Functional Responses between FAW and CBW

Both FAW which preyed on CBW and CBW which preyed on FAW (Figure 1) yielded positive $P_1$ values, indicating that the response of either species to the prey density was in line with the Holling type III functional response (Table 2). The predated proportion increased first, and then decreased as the prey density increased, which is a typical feature of the type III predation functional response (Figure 2). Based on the predation handling time ($T_h$), attack rate ($a$), and theoretical maximum predation value ($N_m$) calculated using the Holling III equation for FAW and CBW (Table 3), the theoretical maximum predation value of the predator decreased with the increasing life stage of the prey. In addition, the highest theoretical maximum predation value of FAW (71.43) was obviously higher than that of CBW (38.46).

**Table 1.** Predation rates (%) of *Spodoptera frugiperda* or *Helicoverpa armigera* on different instar larvae of the other species.

| Predator Species | Larval Instar | Larval Instar of Prey (*H. armigera/S. frugiperda*) | | | | | |
|---|---|---|---|---|---|---|---|
| | | **1st** | **2nd** | **3rd** | **4th** | **5th** | **6th** |
| *S. frugiperda* | 1st | 0 | 0 | 0 | 0 | 0 | 0 |
| | 2nd | 53.3 ± 9.1 Abc | 10.0 ± 5.5 Bc | 0 | 0 | 0 | 0 |
| | 3rd | 76.7 ± 7.7 Ab | 50.0 ± 9.1 Bb | 16.7 ± 6.8 Cd | 0 | 0 | 0 |
| | 4th | 93.3 ± 4.6 Aab | 96.7 ± 3.3 Aa | 50.0 ± 9.1 Bc | 30.0 ± 8.4 Cb | 3.3 ± 3.3 Dc | 0 |
| | 5th | 96.7 ± 3.3 Aa | 96.7 ± 3.3 Aa | 86.7 ± 6.2 Abb | 73.3 ± 8.1 Ba | 20.0 ± 7.3 Cb | 3.3 ± 3.3 Da |
| | 6th | 96.7 ± 3.3 Aa | 100.0 ± 0.0 Aa | 100.0 ± 0.0 Aa | 76.7 ± 7.7 Ba | 76.7 ± 7.7 Ba | 16.7 ± 6.8 Ca |
| *H. armigera* | 1st | 0 | 0 | 0 | 0 | 0 | 0 |
| | 2nd | 36.7 ± 8.8 Ab | 10.0 ± 5.5 Bc | 0 | 0 | 0 | 0 |
| | 3rd | 93.3 ± 4.6 Aa | 66.7 ± 8.6 Bb | 10.0 ± 5.5 Cc | 0 | 0 | 0 |
| | 4th | 100.0 ± 0.0 Aa | 90.0 ± 5.5 Aa | 50.0 ± 9.1 Bb | 3.3 ± 3.3 Cb | 0 | 0 |
| | 5th | 100.0 ± 0.0 Aa | 96.7 ± 3.3 Aa | 63.3 ± 8.8 Bb | 66.7 ± 8.6 Ba | 10.0 ± 5.5 Cb | 0 |
| | 6th | 100.0 ± 0.0 Aa | 100.0 ± 0.0 Aa | 100.0 ± 0.0 Aa | 86.7 ± 6.2 Ba | 56.7 ± 9.1 Ca | 6.7 ± 4.6 D |

Different uppercase letters indicate a significant difference in predation rate by a larval instar of the predator for the different larval instars of the prey, and different lowercase letters indicate a significant difference among different larval instars of the predator for the same larval instar of prey ($\chi^2$ test; $p < 0.05$).

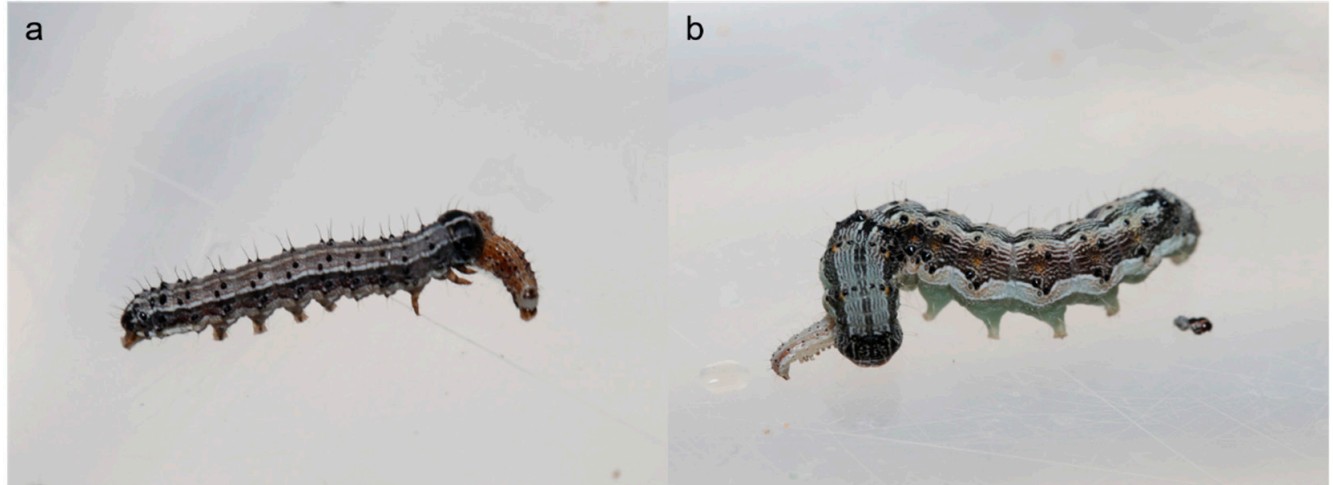

**Figure 1.** Photographs to show the bidirectional predation activity between *Spodoptera frugiperda* (FAW) and *Helicoverpa armigera* (CBW) larvae. (**a**) Sixth-instar larva of FAW preying on CBW. (**b**) Sixth-instar larva of CBW preying on FAW.

**Table 2.** Logistic regression describing the proportion of sixth-instar predator larvae (*Spodoptera frugiperda* or *Helicoverpa armigera*) preying on first–second-instar larvae of the other species.

| Predator Species | Larval Stage of Prey | Parameter | Estimated Value | SE | *T*-Value | Pr(>\|z\|) |
|---|---|---|---|---|---|---|
| *S. frugiperda* | 1st-instar of *H. armigera* | $P_0$ | −0.1382 | 0.2568 | −5.3820 | 0.0000 |
| | | $P_1$ | 0.0896 | 0.0332 | 2.6950 | 0.0122 |
| | | $P_2$ | −0.0025 | 0.0011 | −2.3580 | 0.0262 |
| | | $P_3$ | 0.0000 | 0.0000 | 2.1670 | 0.0396 |
| | 2nd-instar of *H. armigera* | $P_0$ | −1.9691 | 0.7408 | −2.6580 | 0.0133 |
| | | $P_1$ | 0.3781 | 0.1703 | 2.2200 | 0.0354 |
| | | $P_2$ | −0.0215 | 0.0104 | −2.0630 | 0.0492 |
| | | $P_3$ | 0.0003 | 0.0002 | 1.8880 | 0.0702 |

**Table 2.** *Cont.*

| Predator Species | Larval Stage of Prey | Parameter | Estimated Value | SE | *T*-Value | Pr(>|z|) |
|---|---|---|---|---|---|---|
| *H. armigera* | 1st-instar of | $P_0$ | −0.1370 | 0.3875 | −3.5360 | 0.0020 |
| | *S. frugiperda* | $P_1$ | 0.1856 | 0.0828 | 2.2420 | 0.0359 |
| | | $P_2$ | −0.0012 | 0.0047 | −2.6260 | 0.0157 |
| | | $P_3$ | 0.0001 | 0.0000 | 2.6730 | 0.0143 |
| | 2nd-instar of | $P_0$ | −1.7780 | 0.2440 | −7.2860 | 0.0000 |
| | *S. frugiperda* | $P_1$ | 0.1036 | 0.0479 | 2.1640 | 0.0406 |
| | | $P_2$ | −0.0027 | 0.0021 | −1.2830 | 0.2116 |
| | | $P_3$ | 0.0000 | 0.0000 | 1.1180 | 0.2748 |

$P_0$, $P_1$, $P_2$, and $P_3$ are the intercept, linear, quadratic, and cubic coefficients, respectively, of the logistic regression analysis equation for predators consuming different larval stages of prey.

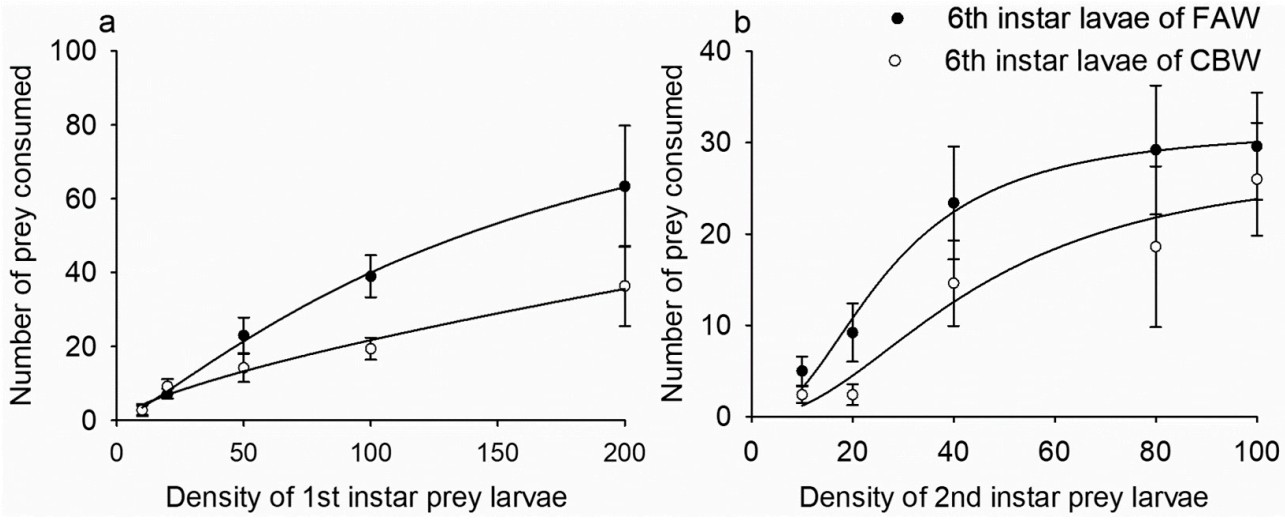

**Figure 2.** Functional response curves of sixth-instar predator larvae of *Spodoptera frugiperda* (FAW) or *Helicoverpa armigera* (CBW) preying on first- (**a**), second (**b**) larvae of each other.

**Table 3.** The attack rate (a), handling time ($T_h$), and theoretical maximum prey consumption (Nm) of sixth-instar larvae of *Spodoptera frugiperda* (FAW) preying on *Helicoverpa armigera* (CBW) and CBW preying on FAW at different prey stages.

| Predator | Stage of Prey Species | Model | *a* | $T_h$ (d) | $N_m$ | $R^2$ |
|---|---|---|---|---|---|---|
| FAW | 1st-instar of CBW | III | $b_1 N_0$ | 0.014 ± 0.001 | 71.43 | 0.871 |
| | 2nd-instar of CBW | III | $b_2 N_0$ | 0.031 ± 0.002 | 32.26 | 0.818 |
| CBW | 1st-instar of FAW | III | $b_3 N_0$ | 0.026 ± 0.003 | 38.46 | 0.732 |
| | 2nd-instar of FAW | III | $b_4 N_0$ | 0.035 ± 0.005 | 28.57 | 0.795 |

In the best-fit type III model, $a = bN_0$, $b_1 = 0.011 ± 0.002$, $b_2 = 0.042 ± 0.010$, $b_3 = 0.007 ± 0.002$, $b_4 = 0.014 ± 0.005$, and $N_0$ = initial density of prey.

### 3.3. Interspecific Competition between FAW and CBW on Maize Plants

When the first-instar larvae of FAW and CBW co-infested maize tassels, the survival rates were 75.6% and 57.8%, respectively, with no significant differences ($F_{1,4} = 5.815$, $p = 0.073$). When the age of the two pests differed (the first-instar larva of FAW with third-instar larva of CBW; first-instar larva of CBW with third-instar larva of FAW), the older pest had a significantly higher survival rate than the younger (third-instar FAW larvae: $F_{1,4} = 62.515$, $p = 0.001$; third CBW: $F_{1,4} = 35.591$, $p = 0.004$), but the survival rate of the first-instar FAW larvae (44.4%) was higher than that of CBW (23.3%) (Figure 3), indicating that the FAW larvae had a greater survivability.

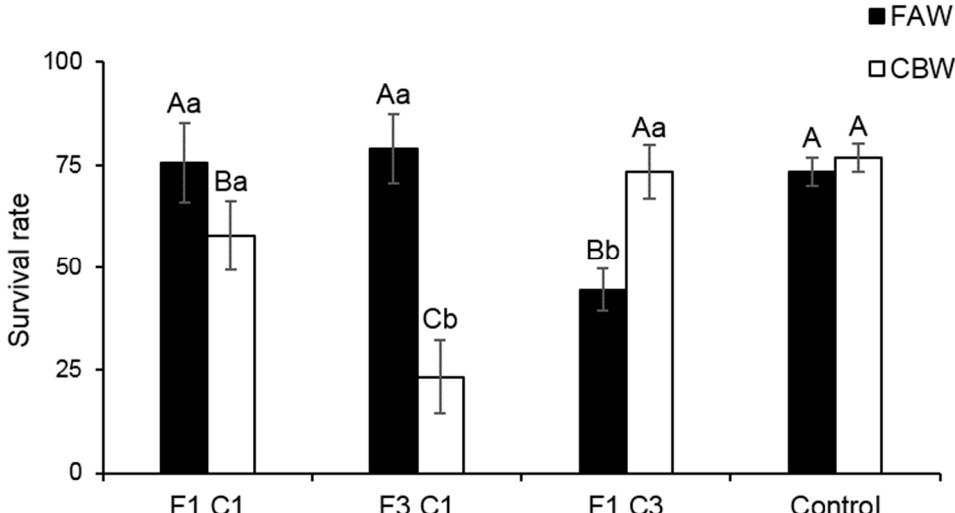

**Figure 3.** Survival rates after a 5-day co-infestation of *Spodoptera frugiperda* (FAW; F1, first-instar; F3, third-instar) and *Helicoverpa armigera* (CBW; C1, first-instar; C3, third-instar) larvae when paired at different stages. Different capital letters above error bars indicate significant differences in the survival rate of FAW or CBW among different treatments; different lowercase letters indicate significant differences in the survival rate of FAW or CBW among the same treatment.

### 3.4. Population Abundances of FAW and CBW in Local Maize Fields

Significantly more adults of FAW were trapped in 2021 (21.3) than in 2019 (7.3) and 2020 (4.3) ($F_{2,6} = 13.232$, $p = 0.006$), and similar results were found for CBW (2019: 7.3, 2020: 9.3; 2021: 27.0) ($F_{2,6} = 90.371$, $p = 0.000$). However, the number of FAW adults compared with CBW trapped each year did not differ significantly (2019: $F_{1,4} = 0.000$, $p = 1.000$; 2020: $F_{1,4} = 6.429$, $p = 0.064$; 2021: $F_{1,4} = 1.877$, $p = 0.243$; Figure 4a). On maize at the VT–R1 stage, the number of FAW larvae per 100 maize plants was significantly higher than that of CBW each year (2019: $F_{1,28} = 60.982$, $p = 0.000$; 2020: $F_{1,28} = 21.815$, $p = 0.000$; 2021: $F_{1,4} = 12.213$, $p = 0.001$). In addition, the number of FAW larvae per 100 maize plants in 2021 was significantly higher than in 2019 and 2020 ($F_{2,42} = 7.118$, $p = 0.002$; Figure 4b).

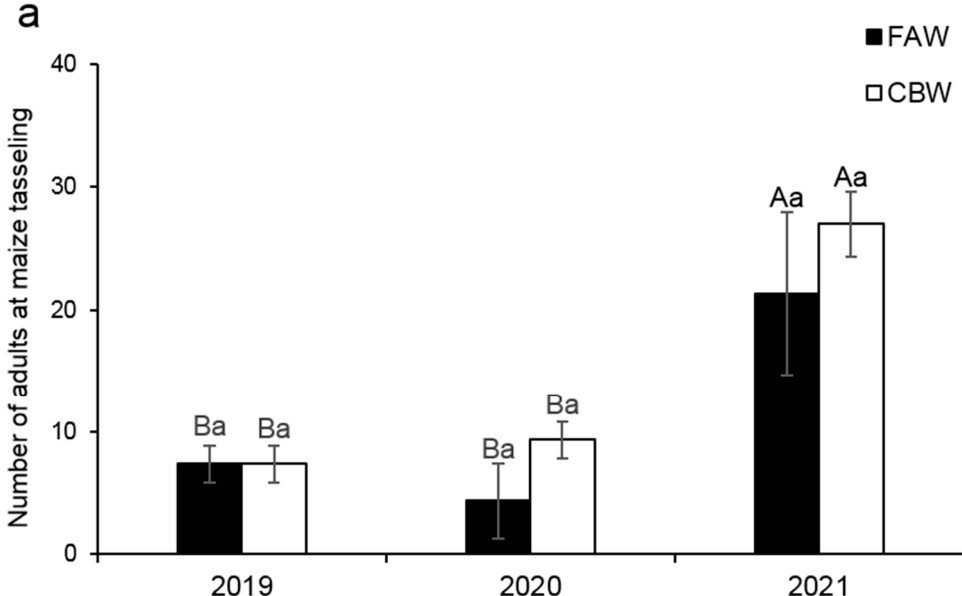

**Figure 4.** *Cont.*

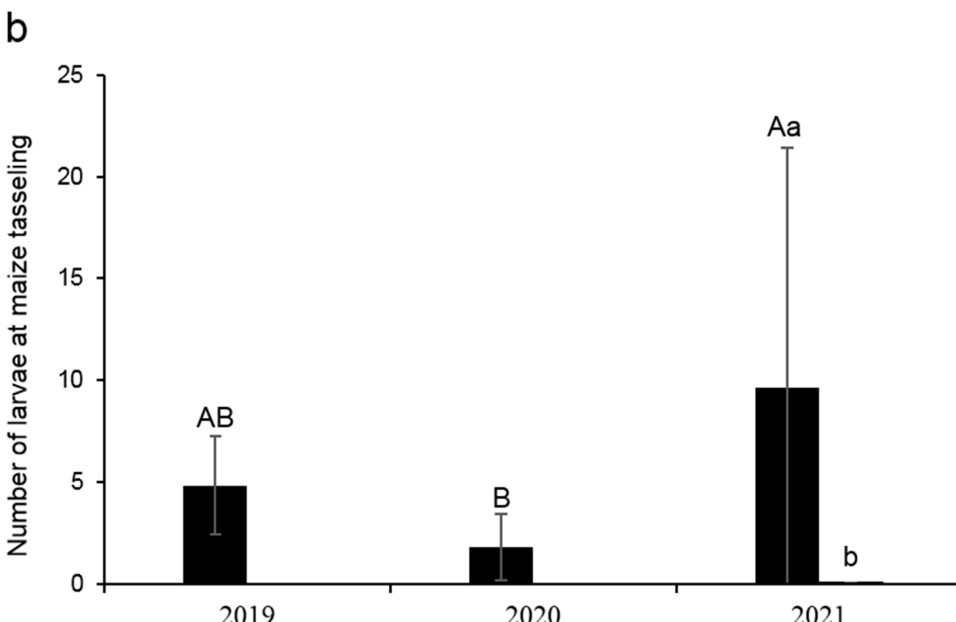

**Figure 4.** Mean number (±SE) of *Spodoptera frugiperda* (FAW) and *Helicoverpa armigera* (CBW) adults (**a**) or larvae (**b**) on maize at the tasseling stages in 2019–2021. Different capital letters above error bars indicate significant differences in the number of FAW or CBW among years; different lowercase letters indicate significant differences in the number of adults or larvae between FAW and CBW in the same year.

## 4. Discussion

Larvae of some lepidopteran pests are known as cannibals [34–37] and can also attack or prey on some natural enemy insects [38–41]. In this study, our laboratory predation assays showed that fall armyworm (FAW) and cotton bollworm (CBW) larvae preyed on each other, and the older larvae had a higher predatory ability than young larvae. However, the predation ability and survivability of FAW larvae were higher than for CBW, as shown by their functional responses at the same developmental stage. A similar phenomenon was also observed when FAW and CBW co-infested the same maize plants in the field trials.

Insect behavior plays an important role in interspecific competition [42]. Several studies have confirmed that FAW has a competitive advantage over several other lepidopteran pests (e.g., *Helicoverpa zea*) in their original habitat due to FAW having superb predatory and survival skills [43–46]. As an invasive species in China, the current study showed that FAW has a high predatory ability in comparison with the native herbivores, CBW, in the predation response assays and co-infestation field trials. In China, Zhao et al. showed that FAW larvae carry out fewer lethal attacks (strikes) and more defensive movements than do the larvae of *Ostrinia furnacalis* (Guenée), which then demonstrated a stronger competitive advantage in the local maize field [46]. Moreover, during interspecific interactions among maize-inhabiting species, FAW is much more aggressive and has a higher survival rate than native *S. litura* [28].

Body size (life stage) also plays a core role during intense interspecific (or intraspecific) competition, as seen in some arthropods such as certain spiders or immature lepidopterans [47]. Several studies have demonstrated that the latter instar larvae of FAW have a stronger predatory ability. Bentivenha et al. (2017) have shown that FAW had greater survival when competing with *Helicoverpa* spp. in several scenarios where larvae were the same (or larger) instar as their competitor, and the influence of interaction on their behavior was also compared [43]. This study further showed the predation ability of all larval stages between FAW and CBW by bidirectional predation; specifically, we used functional response to test the predation ability of the older larvae. Furthermore, we also confirmed the influence of interference competition in wild populations. Usually, once larvae reached

the third instar, FAW or CBW exhibited aggressive and cannibalistic behavior [48,49]. In predation functional responses, only a few of the second-instar larvae of FAW (0 to 1.4%) or CBW (0 to 5.7%) were cannibalized by conspecifics. Thus, the effects of cannibalism on the functional response (especially for the first–second-instar larvae prey) can largely be ignored. Similarly, in the bidirectional predation of FAW and the hoverfly *Episyrphus balteatus* or *E. corollae*, FAW is preyed on by hoverflies when its larvae instar is lower than the third instar, but the FAW fifth- and sixth-instar larvae could prey on hoverflies [40,41].

Temperature and rainfall are the main environmental factors affecting the occurrence of CBW; e.g., excessive rainfall will lead to the death of pupae in the soil [20,50]. In recent decades, the rising temperatures and decreasing precipitation in Yunnan Province facilitated the occurrence of CBW on wheat, rape, pepper, tomato, and pomegranate [51–54]. The number of daily trapped CBW and FAW adults increased from 2019 to 2021, but the infestation level of CBW larvae remained quite low in maize fields, possibly reflecting a predominant advantage of FAW over CBW in maize fields, e.g., FAW suppressed the CBW population via predation. Although the field population was influenced by several other factors, such as pesticide management and climate, the population dynamics at the same time and space still partly reflect the results of the interactions. This was also confirmed by our field trials, which showed that first-instar FAW larvae survived more than that of CBW when it co-infested the same plant with first-instar or third-instar CBW (FAW) larvae. In addition, FAW readily develops on maize plants of varying phenological stages, while CBW primarily consumes its reproductive growth stages [20,26]. Thus, early-instar CBW larvae risk facing FAW late-instar larvae, leading to a low survival rate of CBW. Song et al. (2021) observed that the early development stages of FAW could repel *S. litura* larvae from maize whorls [28]. In Uganda, resident stemborers *B. fusca* and *Chilo partellus* were possibly displaced from maize to sorghum to evade competition from invasive FAW larvae [55,56]. The prevalence of CBW in other host plants requires more extensive and long-term investigations in the future, as acquiring this information is important in developing pest monitoring and management strategies.

The invasion of FAW has caused great economic losses in crop production and heightened pesticide application, which threatens the environment and natural enemies in maize fields and has begun to exert a profound impact on the agricultural ecosystems of China [57]. In view of these challenges, regional control strategies and a national platform for FAW should be implemented using integrated pest management to reduce pest risk and impact [27,28]. In southern China, heightened population levels of FAW in local maize crops can lead to an increased abundance of natural enemies that favor biological control in nearby (non-maize) crops, thereby reducing the need for chemical pest management. Genetically modified *Bacillus thuringiensis* (Bt) crop varieties have permitted an area-wide suppression of polyphagous herbivores such as CBW [58]. A recent study also showed that Bt maize could efficiently suppress the FAW population without the use of chemicals [59]. The deployment of Bt crops can bolster the ecological regulation of pest populations over extensive areas [60]. Based on the data from the long-term follow-up survey, the first generation of CBW damages wheat, whereas successive generations alternate between cotton and maize in northern China, [20]. Area-wide planting of Bt cotton in China has suppressed CBW populations on multiple hosts, including cotton, maize, and other crops [58]. However, an adjustment of the cropping structure, e.g., decreasing the cotton area, has increased the prevalence of CBW on maize since 2010 [61]. According to data from the Chinese National Agricultural Technology Extension Service Center, the occurrence area of CBW in maize fields has increased from 5.59 million hectares in 2017 to 5.70 million hectares in 2020, respectively, while FAW only occurred in 1.35 million hectares in 2020 [62,63]. The current low FAW prevalence in northern China might be caused by the fact that FAW can only survive winter in an annual breeding area south of the January 10 °C isotherm; the effort in controlling FAW in this region suppressed the migratory populations migrating to the northern region. However, in view of the stronger interference competitiveness, FAW

should be considered when constructing a regional maize pest monitoring and management strategy in northern China.

**Author Contributions:** K.W. conceived and designed the research; Y.S. conducted the experiments; Y.S., H.L., H.Z. and K.W. analyzed the data; Y.S., H.L., L.H., S.Z., X.Y. and K.W. wrote the manuscript. All authors have read and agreed to the published version of the manuscript.

**Funding:** This research was funded by China Agriculture Research System (CARS-02).

**Data Availability Statement:** The datasets used and/or analyzed during the current study are available from the corresponding author upon reasonable request.

**Conflicts of Interest:** The authors declare no conflict of interest.

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
