# Peer review of "Interspecific Competition between Invasive Spodoptera frugiperda and Indigenous Helicoverpa armigera in Maize Fields of China"

_agronomy, doi:10.3390/agronomy13030911_

Round 1

Reviewer 1 Report

The topic of the manuscript is relevant and interesting. A huge experimental work is described in understandable language. However, the authors should work on the presentation of the material. The following is a partial list of the shortcomings of this manuscript.

Line 63: The phrase “of their larvae” seems redundant in this sentence.

There is confusion over the years in part “2.1 Insects and maize plants”: Experiments were conducted in 2019–2021. Later instar larvae of FAW were collected in June 2020. CBW colony used in this study was collected in 2021 (in what month?). Both FAW and CBW colonies were reared for around 1 year after collection in the field. Maize was planted in December 2021. – It is not clear for what purpose maize was planted at the end of 2021, if the experiments were carried out in 2019-2021. How many months did the laboratory experiment using two Noctuidae species of different instar last?

Line 102: Freshly hatched predator larvae – In this sentence, it should be clarified again about freshly molted 6th instar larvae.

Line107: Please specify how many predators (6th instar larvae) of each species were used in this experiment.

Line 184. Think about the title of the table 1. The title should match the content of the table. It looks like the authors have merged the two tables but left the name of the first one.

Line 233, 234, 249: rate of FAW or of CBW – remove the second “of”

Line 238: Probably 2022 year is listed by mistake.

Line 329: …hosts including cotton, maize and other host crops – It's better to remove word "host" before "crops".

Replace the literal designations A, B with a, b in Figure 1, as in other figures. Photos look unnatural, it would be better to replace them with brighter ones.

Author Response

Comments to the Author

The topic of the manuscript is relevant and interesting. A huge experimental work is described in understandable language. However, the authors should work on the presentation of the material.

Response: We are grateful for your valuable comments. Here, we revised the relevant description in the materials and methods, e.g., the trial time, collection area of pests, purpose of maize plants and details in the laboratory assays. 

Line 63: The phrase “of their larvae” seems redundant in this sentence.

Response: The phrase “ of their larvae” has been deleted. (Line 62)

There is confusion over the years in part “2.1 Insects and maize plants”: Experiments were conducted in 2019–2021.

Response: Both laboratory experiment and field co-infestation trials began in December 2021, and pest occurrence data in local maize fields was collected in 2019-2021. Now, we have deleted “Experiments were conducted in the village of Mengmao in 2019–2021” to avoid confusion. (Lines 66).

Later instar larvae of FAW were collected in June 2020. CBW colony used in this study was collected in 2021 (in what moth)

Response: It has been revised to “Later instar larvae of FAW were collected from maize fields in Mengmao (23.58°N, 97.48°E), Ruili City, Yunnan Province in June 2020. CBW old larvae were collected from maize fields in Langfang (39°51´N, 116°60´E), Hebei Province in July 2020, and reared in Mengmao in March 2021.” (Lines 66-69)

Maize was planted in December 2021. – It is not clear for what purpose maize was planted at the end of 2021, if the experiments were carried out in 2019-2021.

Response: Revised. We added the purpose of maize plants: “Maize plants (growing to R1 stage) were taken as food source of the insect in the laboratory assays and conducted co-infestation trial in field”. (Lines 80-81)

How many months did the laboratory experiment using two Noctuidae species of different instar last?

Response: The laboratory experiment lasted nearly two months. We have added this information in Lines 93-94.

Line 102: Freshly hatched predator larvae – In this sentence, it should be clarified again about freshly molted 6th instar larvae.

Response: It has been revised to “Freshly molted (<12–h) 6th instar larvae predator were starved for 12 h before testing”.(Lines 102-103)

Line 107: Please specify how many predators (6th instar larvae) of each species were used in this experiment.

Response: We have clarified this statement as “Each predator–prey combination (one single 6th instar larva predator vs varying number of 1st-2nd instar larvae of prey) was replicated five times”. (Lines 107-108)

Line 184. Think about the title of the table 1. The title should match the content of the table. It looks like the authors have merged the two tables but left the name of the first one.

Response: We are sorry for this mistake. Here, the title has been revised to “Predation rates (%) of Spodoptera frugiperda or Helicoverpa armigera on different instar larvae of the other species”.(Line 184-185)

Line 233, 234, 249: rate of FAW or of CBW – remove the second “of”

Response: Revised. (Line 230, 231, 245)

Line 238: Probably 2022 year is listed by mistake.

Response: We have changed 2022 to 2020. (Line 234)

Line 329: …hosts including cotton, maize and other host crops – It's better to remove word "host" before "crops".

Response: Revised. (Line 336)

Replace the literal designations A, B with a, b in Figure 1, as in other figures. Photos look unnatural, it would be better to replace them with brighter ones.

Response: Revised. (Line 210)

Reviewer 2 Report

Review of “Interspecific competition between invasive Spodoptera frugiperda and indigenous Helicoverpa armigera in maize fields of China”

This paper is quite detailed in its methods of studying these two lepidopteran pests.

While it is interesting to know that they will predate each other, I think the authors should include notes on whether the two species are cannibalistic and will devour their own larvae.

So do the larvae prey on themselves as well as each other?

Also can the authors provide a list of other predators and parasites in the maize fields and note whether they attacked both lepidoptera species equally.  Did fungal parasites favor FAW over CBW or vice versa?  Did bacterial, protozoan or viral diseases appear more frequently on FAW or CBW?  

Did the authors note any vertebrate predators of various stages of these moths, like birds, mice, lizards, etc. 

So while the tests that the authors performed are a good beginning, they are incomplete regarding the total picture of the ecology and interactions of these two moth species.    

Author Response

While it is interesting to know that they will predate each other, I think the authors should include notes on whether the two species are cannibalistic and will devour their own larvae.

So do the larvae prey on themselves as well as each other?

Response: We are very grateful for your valuable comments. Usually, once larvae reached 3rd instar, FAW or CBW exhibited aggressive and cannibalistic behavior. In predation functional responses, we set prey feeding on maize tassels without any predators served as control to checked the impact of cannibalism on the experiment and showed that only a few 2nd instar of FAW (0 to 1.4%) or CBW (0 to 5.7%) larvae were cannibalized by conspecifics. Thus, the effects of cannibalism on the predation functional response (especially for the 1st-2nd instar larvae prey) can largely be ignored. (Lines 277-282)

Also can the authors provide a list of other predators and parasites in the maize fields and note whether they attacked both lepidoptera species equally.

Response: Accepted. It is a very interesting topic considering predators and parasites in the scenario of interspecific competition between FAW and CBW. In the field trials of interspecific competition between FAW and CBW on maize plants, control treatments included only one 1st instar larvae of FAW or CBW on maize tassels were included, but survival rate of FAW (~75%) and CBW (~75%) 5 days after infestation was quite similar, suggesting predators and parasites did not attack FAW or CBW over the other one. However, we have added a list of predators and parasites to FAW and CBW in Discussion and they might play important roles in the interspecific competition between these two species and should be studied further. (Lines 318-320)

Did fungal parasites favor FAW over CBW or vice versa? Did bacterial, protozoan or viral diseases appear more frequently on FAW or CBW?

Response: Revised. In Discussion, we added a paragraph about pathogens: “Some virus occurred through cannibalism of infected larvae, e.g., iridovirus and granulosis virus. Cannibalism in the presence of virus-infected conspecifics was highly cost to FAW, due to the low vitality and more likely to be victims [57]. The similar phenomenon was observed in the cannibalism of CBW with nuclearpolyhedrosis virus [58]. However, the transmission of viruses or other pathogens between FAW and different species through intraguild predation are few reported, thus further research is necessary. Notably, cannibalism of FAW larvae may correlates with its success in interference competition with native stemborer [59].” (Line 307-314)

Did the authors note any vertebrate predators of various stages of these moths, like birds, mice, lizards, etc. So while the tests that the authors performed are a good beginning, they are incomplete regarding the total picture of the ecology and interactions of these two moth species.

Response: Revised. We added relevant discussions as follow: “The invasion of FAW has caused great economic losses of crop production and heightens pesticide application, which threatened the environment and natural enemy in maize fields and posed profoundly impact on agricultural ecosystems of China [62]. FAW has rich parasitic (121 species of parasitoids, 66 species of tachinids) and predatory (Ladybugidae, Cariaidae, etc.) natural enemies worldwide, and has a great application prospect [63]. Although vertebrate predator such as birds and bats also play an important role in controlling FAW on some farms, while the lack of enough case studies from maize makes it challengeable to assess the mechanism and effectiveness of these animal predation on FAW or CBW [64]. Therefore, further research on the role of vertebrate predators in controlling pests is warranted.” (Line 318-324)

Reviewer 3 Report

The paper presents highly interesting data on experiments on the inter-specific competition between FAW and CBW. The paper is well organized, and the authors have analyzed the data and developed predation models of the two species-stage specific interactions.

I sympathize with the authors having to write in a second language (I know no Mandarin), but to publish in an English language journal, the text must be upgraded. The summary was readable and suggests the remaining text should be similarly upgraded. In the summary, they conclude that FAW is dominant in Chinese maize field due to its advantage over CBW – the data would suggest this but not conclusively. L46 & 53 make similar strong statements.  

The feeding biology of CBW is described in L 57 but not for FAW.

L69-71 The origins of the two species are quite different (16 degrees latitude) -- why?.

L87 Clarity - The Predation assays were performed on the various combinations of species x stage combinations.

L149 brackets are missing from the denominator of the equation.

 I liked the analysis of functional response, but wondered why the Holling model was used and why the emphasis on type II and III. The handling time constant in the model is a fitted constant rather than a measured value. It appeared not to play an important role (Table 3). Would a simpler model have yielded the same results.

Figure 2  The scatter in the data is large – means might be useful – and panel C appear to have a reversal of predator success not explained?

Figure 3 space needed between F3C1, etc.

L238 there appears to be an error – should 2022 be 2020 for CBW

Fig 4 y-axis  -- number of larvae (adults) at maize tasseling stage

Discussion – the text needs to be carefully reviewed for syntax and grammar – e.g., L 295-296  was remained -> remained  reflected ->reflecting           etc.

L307-308 – is this based on data or on observation?

L328-332 – is this based on data or on observation?

Round 2

Reviewer 1 Report

The authors tried to make some changes to improve the quality of the manuscript, but only new shortcomings appeared. It is necessary to correct the entire manuscript, especially its final part.

The meaning of many sentences is not clear. There are a lot of mistakes of different kinds in almost every sentence. I will give just a few examples.

Lines 21-22: 38.5, 28.6 individuals - 38.5 and 28.6 FAW individuals.

Lines 66-69: Later instar larvae – Late instar larvae; old larvae – late instars.

Lines 84-89: petri dish - Petri dish.

Line 94: There is no dot at the end of the sentence.

Line 102-103: Freshly hatched (<12–h) 6th instar larvae predators were starved for 12 h before testing. (hatched – molted).

Line 110: 2.4.1Interspecific - 2.4.1 Interspecific.

Line 249: Larvae of some lepidopteran pests are known as cannibalism (cannibalism – cannibals).

Lines 307-346: The paragraphs are not relevant to this study, i.e. there is no adequate conclusion.

Newly added references highlighted in red could have been used if you had studied pathogens and natural enemies of cutworms.

I believe that the next version of the manuscript will be final, either ready for publication or rejected. I recommend that you carefully check the grammar, style, wording of sentences and write clear conclusion.
